# Acute and chronic effects of a light-activated FGF receptor in keratinocytes in vitro and in mice

Theresa Rauschendorfer[1], Selina Gurri[1], Irina Heggli[1], Luigi Maddaluno[1], Michael Meyer[1], Álvaro Inglés-Prieto[2], Harald Janovjak[2,3,4], Sabine Werner[1]

**FGFs and their high-affinity receptors (FGFRs) play key roles in development, tissue repair, and disease. Because FGFRs bind overlapping sets of ligands, their individual functions cannot be determined using ligand stimulation. Here, we generated a light-activated FGFR2 variant (OptoR2) to selectively activate signaling by the major FGFR in keratinocytes. Illumination of OptoR2-expressing HEK 293T cells activated FGFR signaling with remarkable temporal precision and promoted cell migration and proliferation. In murine and human keratinocytes, OptoR2 activation rapidly induced the classical FGFR signaling pathways and expression of FGF target genes. Surprisingly, multi-level counter-regulation occurred in keratinocytes in vitro and in transgenic mice in vivo, including OptoR2 down-regulation and loss of responsiveness to light activation. These results demonstrate unexpected cell type–specific limitations of optogenetic FGFRs in long-term in vitro and in vivo settings and highlight the complex consequences of transferring optogenetic cell signaling tools into their relevant cellular contexts.**

## Introduction

FGFs comprise a group of 22 structurally related proteins in mammals, which play key roles in development, repair, and disease (Beenken & Mohammadi, 2009; Ornitz & Itoh, 2015). Most of them signal through four tyrosine kinase receptors (FGFR1-4). The biological output of FGF signaling depends on the type of FGF and FGFR, on the cell type and on the presence of heparan sulphate proteoglycans on the cell surface, which are required for FGF receptor binding and activation (Beenken & Mohammadi, 2009; Ornitz & Itoh, 2015). For example, FGF7 and FGF10 both activate the "b" splice variant of FGFR2 (FGFR2b), but FGF7 induced receptor degradation and cell proliferation in HeLa cells, whereas FGF10 treatment induced receptor recycling and cell migration (Francavilla et al, 2013).

Little is known on how different cellular responses are activated by stimulation of the same receptor with different FGFs or by stimulation of different receptors with the same ligand. In addition, there are unsolved issues regarding possible unique or synergistic effects of different types of FGFRs. This is particularly relevant for cells, which express multiple FGF receptors, such as keratinocytes. These cells express FGFR2 and FGFR3 and very low levels of FGFR1, with FGFR2 being the most relevant receptor for epidermal function (Yang et al, 2010; Meyer et al, 2012). Their activation is of crucial importance for the maintenance of epidermal barrier function and wound healing, and mice lacking FGFR1 and FGFR2 in keratinocytes develop a phenotype resembling the inflammatory skin disease Atopic Dermatitis (Yang et al, 2010) and show a severe defect in wound re-epithelialization (Meyer et al, 2012).

To test the contribution of an individual FGFR to a cellular function, it should be selectively activated in a precise manner both in space and time, which is not possible using the ligands. A promising alternative is optogenetics, where photoactivatable protein domains are used to tightly control and manipulate cellular signaling networks (Gorostiza & Isacoff, 2008; Airan et al, 2009; Toettcher et al, 2011; Tischer & Weiner, 2014; Pudasaini et al, 2015; Rogers & Muller, 2020). FGFRs were among the first optogenetic tools developed specifically to manipulate cellular signaling. Initially, two versions of a light-regulated FGFR1 were generated and characterized. One version takes advantage of aureochrome light-oxygen-voltage (LOV) domains for receptor homodimerisation, and illumination of this modified FGFR1 activated the canonical intracellular signaling cascades and induced proliferation, migration and in vitro capillary formation of endothelial cells (Grusch et al, 2014). The second uses cryptochrome 2 homo-interaction to induce receptor clustering, and allowed light-induced activation of downstream signaling pathways, cytoskeletal reorganisation, and directed cell migration (Kim et al, 2014). Further modifications of FGFR1 rendered it sensitive to red light (Reichhart et al, 2016; Leopold et al, 2019, Leopold et al, 2020), induced membrane recruitment of the receptor (Krishnamurthy et al, 2020) or of a photodomain (Bugaj et al, 2013), or resulted in receptor inactivation upon

[1]Department of Biology, Institute of Molecular Health Sciences, Eidgenössische Technische Hochschule (ETH) Zurich, Zurich, Switzerland   [2]Institute of Science and Technology (IST) Austria, Klosterneuburg, Austria   [3]Australian Regenerative Medicine Institute, Faculty of Medicine, Nursing and Health Sciences, Monash University, Clayton, Australia   [4]European Molecular Biology Laboratory Australia, Monash University, Clayton, Australia

Correspondence: Sabine.werner@biol.ethz.ch; Harald.janovjak@monash.edu

illumination (Kainrath et al, 2017; Kainrath & Janovjak, 2020). However, optogenetic FGFR activation was never extended past FGFR1. Furthermore, light-activated FGFRs and light-activated receptors or signaling proteins in general have not been deployed in vivo in transgenic rodents, which would allow precise timing of FGFR activation in a tissue-specific manner if the transgene is driven by specific promoters. Therefore, important open questions exist in the context of transferring this and also other optogenetic cell signaling technology to relevant biological contexts: (1) Are light-activated signaling receptors applicable in transgenic animals (Zeng & Madisen, 2012; Ting & Feng, 2013)? (2) What are potential negative consequences and how do these depend on cell type and strategy? Studies with virally delivered and recombinase-enabled optogenetic actuators (Nectow & Nestler, 2020) or bacterial cyclic nucleotide producing enzymes (Jansen et al, 2015; Luyben et al, 2020) or experiments in transgenic invertebrate models (Husson et al, 2013; Guglielmi et al, 2015; Johnson et al, 2017; Bunnag et al, 2020) have shown promising results. However, the general feasibility of non-neuronal optogenetics, which uses this strategy to activate cell signaling in contexts outside of the nervous system, has not been demonstrated in transgenic rodents.

Because of the important function of FGFR2 in keratinocytes, we generated a version of this receptor, which gets activated by blue light instead of its natural ligand (named OptoR2). OptoR2 robustly activated FGFR signaling and induced migration and proliferation in human embryonic kidney (HEK) 293T cells and caused FGFR signaling, alterations in gene expression and short-term migration in keratinocytes. However, down-regulation of the receptor was observed in keratinocytes, suggesting that long-term activation of OptoR2, even at a low level, is deleterious for these cells. These results unravel advantages, but also some limitations of optogenetics for the analysis of growth factor signaling.

# Results

### Light induces FGFR2 activation in OptoR2-expressing HEK 293T cells

We established an optogenetic approach to selectively activate FGFR2 in keratinocytes. For this purpose, an epitope-tagged aureochrome LOV domain was fused C-terminally to the intracellular part of FGFR2. The extracellular ligand binding domains were removed, and the receptor was anchored to the cell membrane using a myristoylation signal (Fig 1A). Consequently, OptoR2 is not responsive to its natural ligands, but only to blue light.

HEK 293T cells, which stably and constitutively express OptoR2, responded to 15-min exposure to blue light with phosphorylation/activation of ERK1/2, which was comparable to the level achieved with EGF (Fig 1B). By contrast, vector-transfected control (ctrl) and OptoR2-expressing HEK 293T cells only responded weakly to FGF7 and FGF10, the ligands of the FGFR2b splice variant that is highly expressed in keratinocytes. This is consistent with the weak expression of FGFR2 in HEK 293T cells compared with immortalized human keratinocytes (HaCaT cells) (Fig S1A). As expected, illumination of ctrl cells had no effect on ERK1/2 activation (Fig 1B).

Immunofluorescence staining confirmed the robust activation of ERK1/2 (Fig 1C).

Using light-emitting diodes (LEDs), we found that OptoR2 was activated by blue light (λ ~ 450 nm) and by white LEDs that emitted light of a continuous spectrum, but not by red light (λ ~ 700 nm) (Fig 1D). This is consistent with the activation characteristics of flavin mononucleotide, the ubiquitously present light-sensitive co-factor of LOV domains. A 1-s illumination pulse with blue light was sufficient to induce ERK1/2 phosphorylation, and 30 s of illumination were required for maximal activation (Fig 1E). Therefore, a remarkable temporal precision can be achieved with OptoR2.

Activation of OptoR2 also resulted in phosphorylation of the scaffold protein FGFR substrate (FRS) 2α (Y196 and Y436), whereas other signaling molecules that are frequently activated by FGFRs, such as AKT and p38 (Ornitz & Itoh, 2015), were neither activated by blue light nor by FGFs (Fig S1B). The kinetics of ERK1/2 activation by OptoR2 was comparable to the activation of FGFR2 by its natural ligands FGF7 and FGF10, although the extent of activation by light was much stronger (Fig S1C).

In scratch wounding assays, OptoR2-expressing HEK 293T cells migrated significantly faster in the light than in the dark, whereas FGF7 and FGF10 had only a weak or no effect, respectively (Fig 1F). Their proliferation was also significantly increased when exposed to blue light for 1 or 24 h, with the longer exposure being more efficient than the effect achieved with FGFs or FBS (Fig 1G). Of note, minimal residual (i.e., in the dark) signaling activity was observed (e.g., Figs 1B and S1B and C), rendering OptoR2 a highly efficient optogenetic tool.

### OptoR2 is functional in primary and immortalized murine keratinocytes, but undergoes down-regulation

We next generated transgenic mice expressing OptoR2 under the control of the keratin 14 (K14) promoter, which targets expression of transgenes to basal keratinocytes of the epidermis and outer root sheath keratinocytes of the hair follicles (Fuchs, 1993). For expression of the transgene, we used a well-characterized expression cassette ((Munz et al, 1999); Fig 2A). Three transgenic mouse lines were obtained, but only primary keratinocytes from one line expressed the transgene at significant levels (transgenic mouse line 1; Figs 2B and S2A). Keratinocytes, which expressed the transgene, responded to 15 min blue light as well as to FGF7 treatment with phosphorylation of ERK1/2 (Fig 2B). No detectable signaling activity of the receptor was observed in the absence of light. These results demonstrate that a promoter of a gene, which is endogenously expressed in keratinocytes, can be used to drive transgene expression. However, we found individual differences in the OptoR2 expression levels between cells from different pups of one litter. Furthermore, freshly isolated primary keratinocytes from the progeny of mouse line 1 (1.5 yr or 3 generations later) no longer responded in the same experimental setting (Fig 2B). This is most likely the consequence of a strong down-regulation of OptoR2, which was observed both at the mRNA and at the protein level (Fig 2C).

We also examined spontaneously immortalized keratinocytes obtained from cells of the first generation of mice. Although these cells showed robust ERK1/2 activation in response to blue light shortly after immortalization, continuous passaging and cultivation for 6 mo completely abolished their light response (Fig 2D). Both illumination and FGF treatment strongly stimulated cell migration

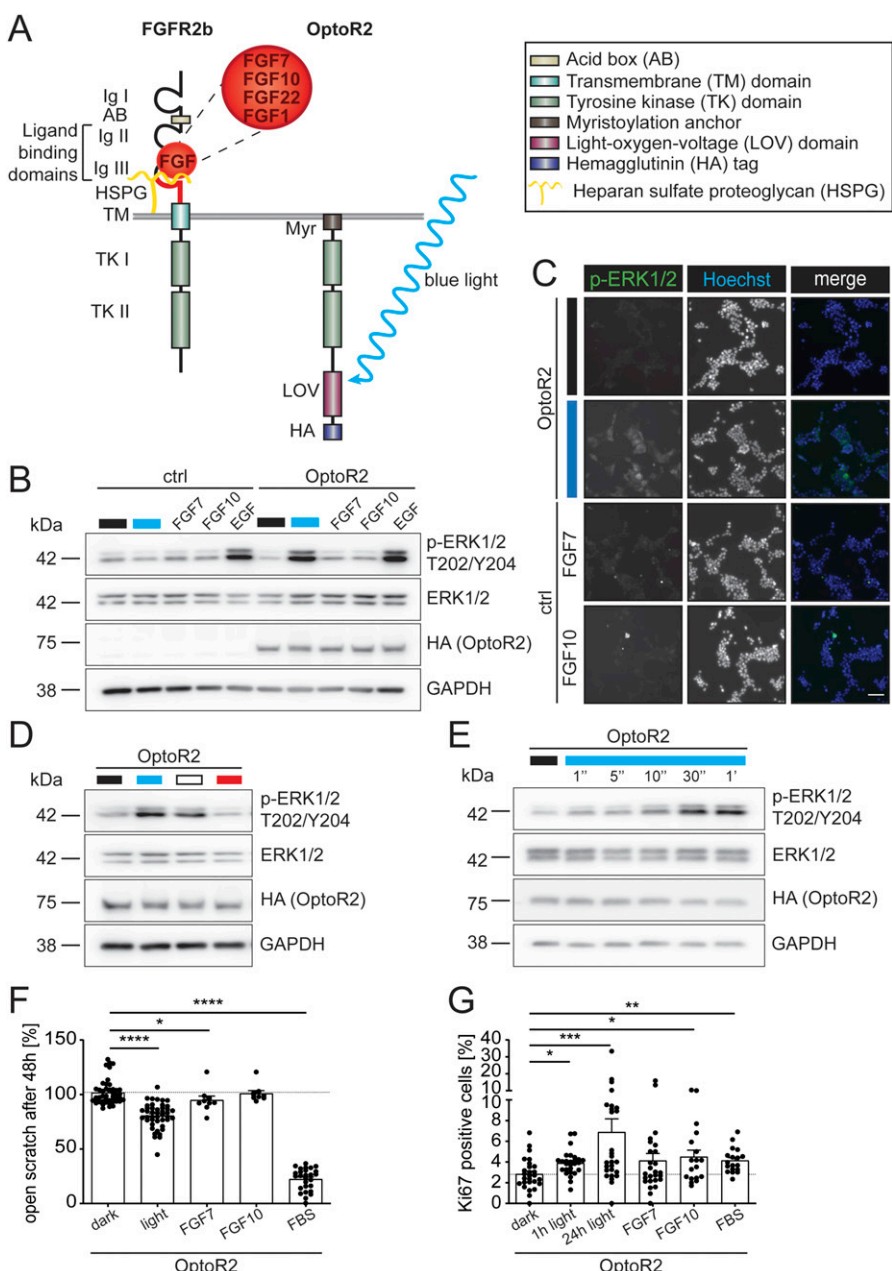

**Figure 1. OptoR2 is functional in HEK 293T cells.**
**(A)** Schematic representation of endogenous FGFR2b and OptoR2. AB, acid box; HSPG, heparan sulphate proteoglycan (HSPG); HA, hemagglutinin epitope; Ig, immunoglobulin-like domain; LOV, light-oxygen-voltage domain; Myr, myristoylation signal; TK, tyrosine kinase domains; TM, transmembrane domain.
**(B, C)** OptoR2 HEK 293T cells and vector-transduced control cells (ctrl) were serum-starved for 16 h. Subsequently, they were left untreated (black bar), or exposed 15 min to blue light (blue bar), or FGF7, FGF10, or EGF (10 ng/ml). **(B)** Western blot for phospho-ERK1/2 (p-ERK T202/Y204), total ERK1/2, HA (OptoR2), and GAPDH (loading control).
**(C)** Immunofluorescence staining for p-ERK1/2 (green). Nuclei were counterstained with Hoechst 33342 (blue). Note that the single channels are shown in black and white; colours are only shown in the merge. Scale bar: 250 μm. **(D, E)** OptoR2 HEK 293T cells were serum-starved for 16 h and then illuminated for 15 min with light of different wavelengths (D) or for 1 s to 1 min with blue light (E). For (E), lysates were prepared 2 min after the onset of illumination. Western blots for p-ERK, total ERK1/2, HA (OptoR2), and GAPDH are shown.
**(F)** OptoR2-transduced HEK 293T cells were cultured in medium with 0.5% FBS for 16 h, subjected to scratch wounding, and illuminated with blue light, or treated with FGF7, FGF10 (10 ng/ml), or 10% FBS. Bar graph shows percentage of open scratch at 48 h. **(G)** OptoR2-transduced HEK 293T cells were serum-starved for 16 h and then illuminated for either 1 or 24 h or treated with FGF7, FGF10 (10 ng/ml), or 10% FBS for 24 h. At 24 h they were analysed by immunofluorescence staining for Ki67. The percentage of Ki67-positive cells is shown. Bar graphs show mean ± SEM. **(B, D, E)** Representative of four experiments. **(C)**: Representative pictures of two experiments. **(F)**: n = 9–45 from three experiments. **(G)** n = 18–27 from two experiments. *P ≤ 0.05, **P ≤ 0.01, ***P ≤ 0.001, ****P ≤ 0.0001 (Mann–Whitney U test). Source data are available for this figure.

at the beginning of the cultivation period, but only FGFs induced migration 6 mo later, whereas light had no effect (Fig 2E). An additional OptoR2 cell line also lost the light responsiveness (Fig S2B).

## Expression of OptoR2 induces skin tumour formation

We also noticed changes in the adult mice of this colony. In the early generations, a high percentage of the transgenic mice, in particular males, developed epithelial skin tumours at light-exposed sites (back, face) at the age of 6–12 wk (Fig 3A, left upper panel), which were highly keratinized and showed strong cell proliferation (Fig 3A, upper right and lower left panels). This phenotype prevented us from performing in vivo illumination experiments, and

the mice were euthanized according to animal welfare regulations at an early stage of tumour development.

Membrane staining for OptoR2 was seen in the tumour cells (Fig 3A, lower right panel), whereas the expression level of OptoR2 in the normal back skin was below the detection limit. The up-regulation of OptoR2 in the tumours is consistent with the strong activity of the K14 promoter in hyperproliferative epithelia (Fuchs, 1993). These findings suggest that the continuous weak activation of OptoR2, which occurs under normal housing conditions, results in tumour formation that is further driven by strong activity of the employed promoter in keratinocytic tumour tissue.

Correlating with our in vitro findings, the tumour incidence strongly declined in the following generations in mice of both sexes

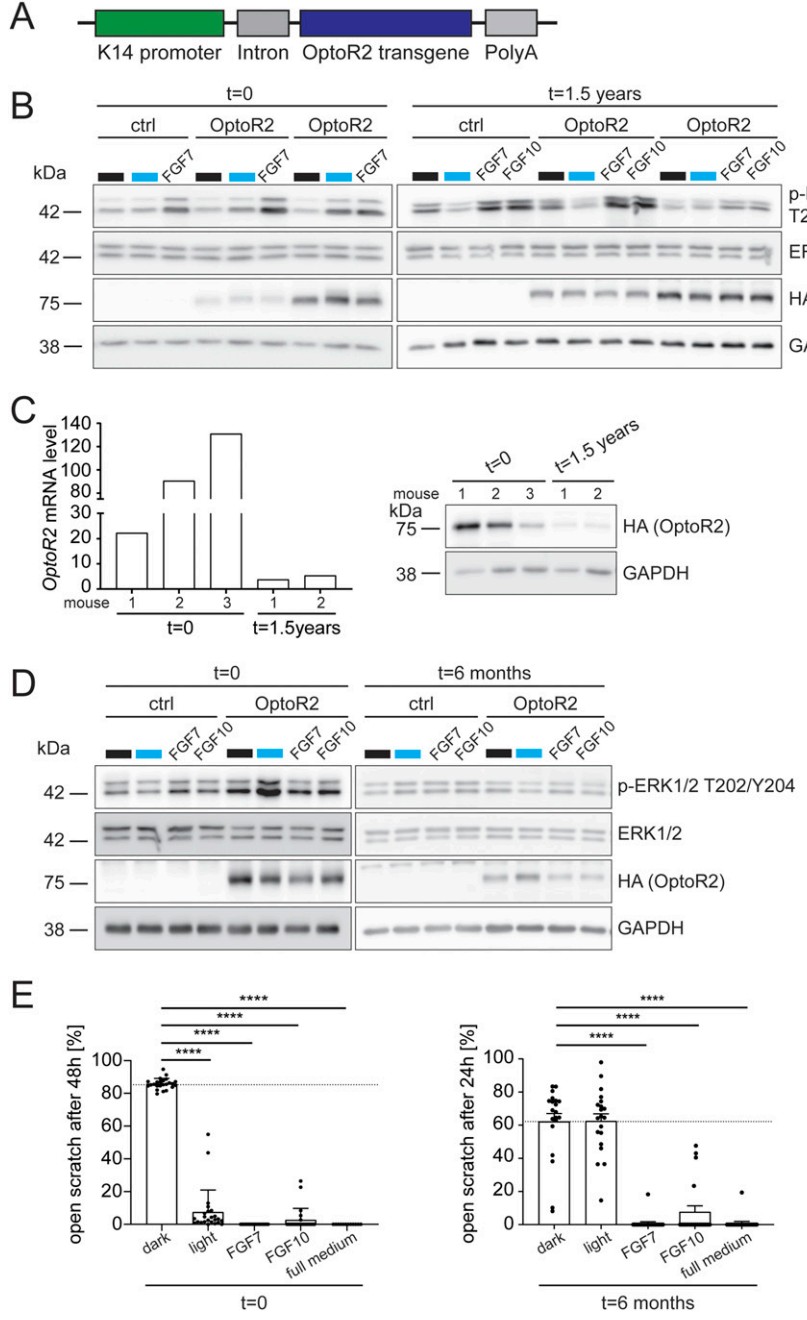

**Figure 2. OptoR2 expression is functional in murine keratinocytes, but down-regulated over time.**
**(A)** Construct used for the generation of the transgenic mice. Functional elements include a human keratin 14 (K14) promoter, a rabbit β-globin intron, the OptoR2 cDNA and the human growth hormone poly A.
**(B, C)** Primary keratinocytes from one newborn ctrl and two OptoR2 mice (transgenic mouse line 1) from the direct offspring of the founder mice or from progeny 1.5 yr (3 generations) later were serum-starved for 24 h and then left untreated or exposed to blue light or FGF7 or FGF10 (10 ng/ml) for 15 min. **(B)** Western blot for p-ERK1/2 (T202/Y204), ERK1/2, HA (OptoR2), and GAPDH. Note that different exposure times were used for the HA blots in the left and right panels. **(C)** Different batches of untreated primary keratinocytes were analysed by qRT-PCR for *OptoR2* relative to *Rps29* and by Western blot for HA (OptoR2) and GAPDH.
**(D)** Immortalized keratinocytes from K14-OptoR2 mice and a wild-type littermate (ctrl) were serum-starved for 24 h and exposed to blue light or FGF7 or FGF10 (10 ng/ml) for 15 min, both at the beginning of the cultivation period (t = 0) and after extended cultivation and passaging (t = 6 mo). Western blots for p-ERK1/2, total ERK1/2, HA (OptoR2), and GAPDH are shown. **(E)** Confluent and serum-starved immortalized keratinocytes were subjected to scratch wounding and exposed to blue light or FGF7, FGF10 (10 ng/ml), or full keratinocyte growth medium. The percentage of open scratch at 24 or 48 h is shown in the bar graphs. Bar graphs show mean ± SEM.
**(B)** Representative of four experiments. t = 0: n = 4 mice per genotype. t = 1.5 yr: n = 12–18 mice per genotype. **(C)** Ctrl littermate expression level was set to one. One experiment, n = 2–3 mice per genotype.
**(D)** Representative of two experiments with three cell lines. **(E)** t = 0: n = 24 from two experiments. t = 6 mo: n = 20 from three experiments. **** ≤ 0.0001 (Mann–Whitney U-test).
Source data are available for this figure.

(Fig 3B). This supports the hypothesis that continuous OptoR2 expression/activation induces counter-regulatory mechanisms both in vitro and in vivo.

### OptoR2 activation by light mimics signaling by endogenous FGFR2 in human keratinocytes

To avoid the counter-regulation in keratinocytes, we generated HaCaT cells expressing OptoR2 under the control of a doxycycline (Dox)-inducible promoter using a lentiviral transduction system (Fig S3A). These cells responded to 15 min blue light illumination or

exposure to FGF7 or FGF10 with phosphorylation of ERK1/2 (Fig 4A), consistent with the presence of OptoR2 at the plasma membrane (Fig 4B).

The kinetics of ERK1/2 activation by light was comparable to the activation of endogenous FGFR2b by FGF7 and FGF10: ERK1/2 was phosphorylated already after 5 min of illumination or FGF treatment, followed by a continuous decrease within 2 h (Fig 4C). This correlated with a rapid (auto)phosphorylation of OptoR2 (Fig 4C). The sensitivity of the pFGFR2 antibody was not sufficient to detect the endogenous pFGFR2 in total keratinocyte lysates (data not shown). FRS2α as well as other downstream effectors were also

# A

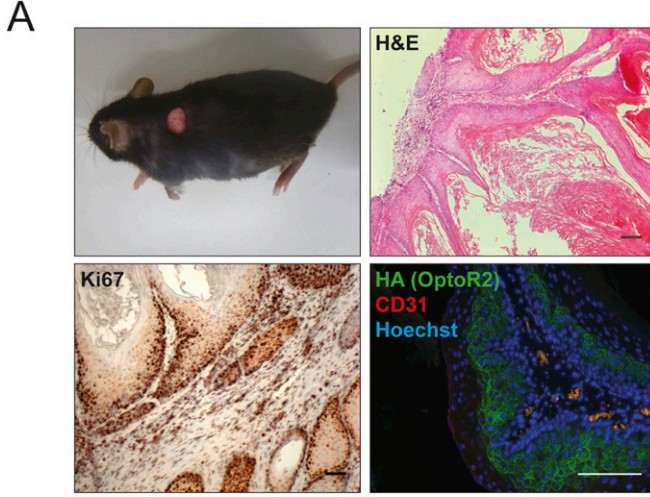

# B

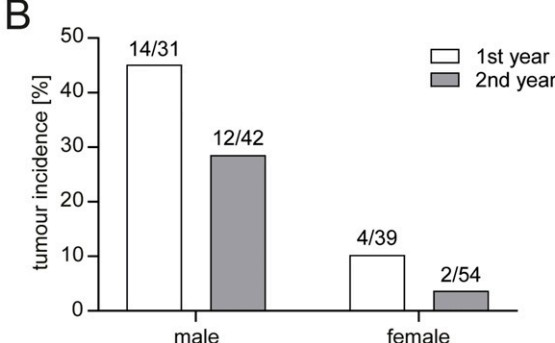

**Figure 3. OptoR2 expression in keratinocytes of transgenic mice induces skin tumour formation, which declines over time.**
**(A)** Left upper panel: Adult K14-OptoR2 mouse with a skin tumour. Other panels: Representative photomicrographs from skin tumour sections; hematoxylin & eosin staining (top right), Ki67 immunohistochemistry and counterstaining with hematoxylin (bottom left), or immunofluorescence staining for HA (OptoR2; green) and the endothelial cell marker CD31 (red) (bottom right). Nuclei were counterstained with Hoechst 33342 (blue). Scale bar: 100 $\mu$m. Representative pictures of n = 11 mice are shown. **(B)** Tumour incidence in adult K14-OptoR2 mice of both sexes during the first and second year after generation of the mouse line. Number of mice with tumours and of total mice is indicated in the graph. Source data are available for this figure.

phosphorylated upon illumination of OptoR2-expressing cells or treatment with FGFs (Fig 4D). We observed no obvious activation of signaling in the dark in this cell line (Fig 4).

In addition, we tested the construct in primary human fibroblasts, which predominantly express FGFR1 (our unpublished RNA sequencing data). In this cell type, all tested signaling effectors were phosphorylated upon illumination of OptoR2 or treatment with FGF2, whereas FGF1, which weakly activates all FGFRs, had only a minor effect (Fig S3B). This result demonstrates the suitability of OptoR2 to activate FGFR2 signaling in various cell types.

### Illumination of OptoR2-expressing keratinocytes activates FGF target gene expression and induces migration

Illumination of the OptoR2-expressing HaCaT cells also activated the expression of known FGFR target genes in keratinocytes,

including dual-specific phosphatase (*DUSP*) 6, heparin-binding EGF-like growth factor (*HBEGF*), and vascular endothelial growth factor (*VEGF*) (Frank et al, 1995; Li et al, 2007; Maddaluno et al, 2020). Their mRNA levels significantly increased after 6 h of stimulation with FGF7, FGF10, or continuous illumination with blue light (Fig 5A). Remarkably, dim light (~30 $\mu$W/cm$^2$) was sufficient for OptoR2 activation. This light intensity is lower than the intensity required to activate many other optogenetic tools, for example, mW/cm$^2$ in the case of engineered ion-conducting opsins (Zeng & Madisen, 2012; Ting & Feng, 2013), demonstrating the high sensitivity of OptoR2. Further reduced light doses through intermittent light pulsing with "1 min on/15 min off" (1/15) or "1 min on/60 min off" (1/60) cycles, which were introduced to prevent possible negative feedback mechanisms of receptor down-regulation over extended illumination periods, were not sufficient to influence the expression of these genes.

The migratory capacity of HaCaT cells in the scratch wounding assay was generally rather weak, even in response to FGF7 or FGF10. Nevertheless, there was a mild increase in migration upon exposure to continuous or 1/60 intermittent light (Figs 5B and S4A). The rather weak effect of light might again be a consequence of receptor down-regulation. Indeed, after 3 h of continuous illumination, OptoR2 protein levels were slightly reduced and after 24 h, OptoR2 was almost undetectable. OptoR2 mRNA also declined with similar kinetics, and this correlated with a reduction in DUSP6 mRNA levels after the initial strong increase, whereas expression of housekeeping genes was not affected (Figs 5C and S4B and C).

## Discussion

We established an optogenetic strategy, which allows induction of FGFR signaling and downstream behavioural responses with remarkable temporal precision in HEK 293T cells. Activation of major FGFR signaling pathways and induction of FGF target gene expression was also achieved in keratinocytes and fibroblasts, demonstrating that OptoR2 is highly suitable to study the early responses to FGFR activation with robust function and minimal residual activity in its natural cellular context. This finding confirms previous data, which showed that expression of endogenous FGFR1 does not affect the activity of an OptoR1 protein that was constructed in the same way as OptoR2 (Grusch et al, 2014). In the future, it will be interesting to express OptoR2 together with OptoR1 and other light-activatable receptors to study individual, overlapping, or synergistic activities of different FGFRs or other receptor tyrosine kinases and to identify signaling molecules and target genes, which are activated by individual receptors.

In spite of these promising results, rapid down-regulation of OptoR2 occurred in murine and human keratinocytes, which was independent of the promoter used to drive transgene expression. These data suggest that mild, but chronic FGFR signaling negatively affects keratinocyte viability or plating efficiency. This would result in selection of cells, which have down-regulated the receptor. We did not observe significant cell death of the OptoR2-expressing keratinocytes at any given time point, but even apoptosis of a small percentage of cells may result in progressive overgrowth by

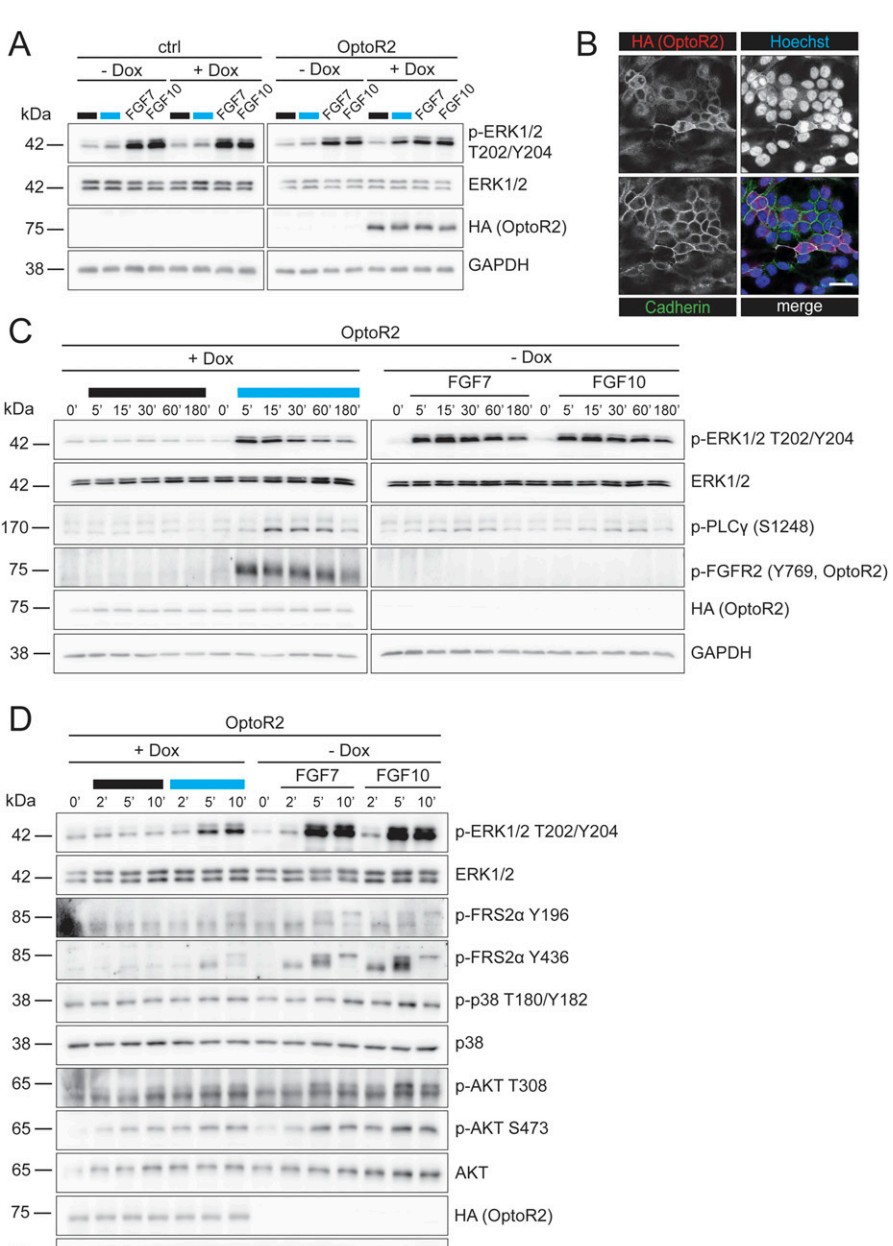

**Figure 4. Activation of inducibly expressed OptoR2 in human keratinocytes mimics FGFR2 signaling.**
HaCaT cells were serum-starved and treated with 50 ng/ml Dox or DMSO (vehicle, -Dox) for 24 h. **(A)** OptoR2- and vector-transduced (ctrl) HaCaT cells were then left untreated, illuminated or treated with FGF7 or FGF10 (10 ng/ml) for 15 min. Western blots for p-ERK1/2, total ERK1/2, HA (OptoR2), and GAPDH. **(B)** Immunofluorescence staining for HA (OptoR2; red) and the membrane protein E-cadherin (green), counterstained with Hoechst 33342 (blue). Single channels are shown in black and white; colours are only shown in the merge. Scale bar: 10 μm. **(C, D)** OptoR2-HaCaT cells were illuminated with blue light or treated with FGF7 or FGF10 (10 ng/ml) for 5, 15, 30, 60, or 180 min (C) or for 2, 5, or 10 min (D). Western blots for OptoR2 (HA), GAPDH, and total and phosphorylated forms of different signaling molecules are shown. **(A)** Representative of six experiments. **(B)** Representative confocal pictures of two experiments. **(C)** Representative of two experiments. **(D)** Representative of six experiments.
Source data are available for this figure.

keratinocytes that have down-regulated the receptor. It is also possible that the differentiation of OptoR2-expressing keratinocytes is enhanced. This hypothesis is supported by the delayed keratinocyte differentiation in mice expressing a dominant-negative FGFR in suprabasal keratinocytes (Werner et al, 1993) and the induction of early and late differentiation in HaCaT keratinocytes by over-expression of FGFR2b and stimulation with FGF7 (Belleudi et al, 2011; Rosato et al, 2018). Although we did not observe a strong effect of OptoR2 expression on keratinocyte differentiation markers in pilot in vitro experiments, a continuous mild effect on differentiation may lead to the loss of some differentiated cells upon passaging.

In the future, it will be interesting to further study the mechanisms underlying the loss of OptoR2 expression or responsiveness in keratinocytes. Epigenetic mechanisms may be involved, as

previously shown for transgene silencing in different mouse lines (Calero-Nieto et al, 2010; Blewitt & Whitelaw, 2013; Gödecke et al, 2017). In addition to loss of transgene expression, negative feedback regulation may occur in response to OptoR2 activation, for example, via activation of endogenous signaling inhibitors (Ornitz & Itoh, 2015). However, this does not explain the rapid loss of the receptor upon continuous illumination of OptoR2-expressing HaCaT keratinocytes. Therefore, active receptor down-regulation is likely to occur as previously suggested for wound keratinocytes, where high levels of FGF7 are present (Marchese et al, 1995). At the protein level, this is achieved by receptor ubiquitination and subsequent internalization, which is frequently followed by proteasomal degradation (Francavilla et al, 2013; Ornitz & Itoh, 2015). Down-regulation of FGFR2 mRNA by FGFR activation has also been observed in fibroblasts

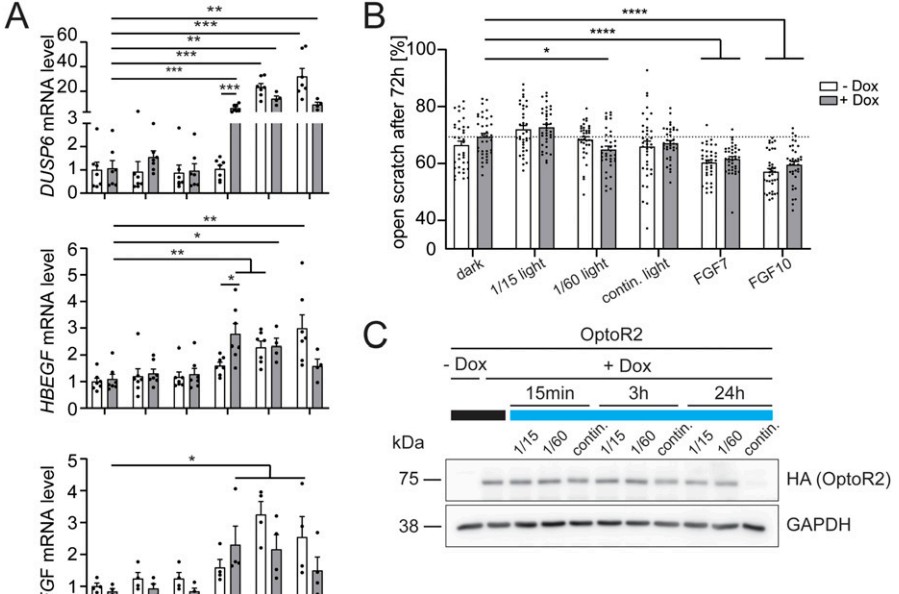

**Figure 5. Activation of OptoR2 in human keratinocytes induces expression of FGF target genes and promotes cell migration.**
OptoR2-transduced HaCaT cells were serum-starved and treated with 50 ng/ml Dox or DMSO for 24 h. **(A)** Cells were left untreated, illuminated with blue light ("1 min on/15 min off" [1/15] cycles, "1 min on/60 min off" [1/60] cycles, continuous illumination) or treated with FGF7 or FGF10 (10 ng/ml) for 6 h. RNA samples were analysed by qRT-PCR for *DUSP6*, *HBEGF*, and *VEGF* relative to *RPL27*. **(B)** Cells were subjected to scratch wounding and illuminated using three different illumination schemes or incubated with 10 ng/ml FGF7 or FGF10. The percentage area of open scratch at 72 h is shown in the bar graph. **(C)** Cells were illuminated for 15 min, 3 h or 24 h using three different illumination schemes and analysed by Western blot for HA (OptoR2) and GAPDH. Bar graphs show mean ± SEM. **(A)** "dark – Dox" expression level was set to one. n = 4–7 from two to five experiments. **(B)** n = 36 from two experiments. **(C)** Representative of three experiments. *$P \leq 0.05$, **$P \leq 0.01$, ***$P \leq 0.001$, **** $\leq 0.0001$ (Mann–Whitney U test).
Source data are available for this figure.

and did not involve reduced transcription (Ali et al, 1995). Therefore, mRNA decay is a more likely mechanism, which may occur in response to FGFR kinase activation and/or continuous light exposure. Our newly developed OptoR2 could be an important tool to address the mechanisms of receptor down-regulation in the future.

There are several possibilities to adjust the optogenetic system to optimize its function. First, the light conditions could be further modified by pulsed illumination (Hennemann et al, 2018) or the expression level of OptoR2 could be lowered by using a different promoter. In addition, the lifetime of LOV-domain photoreceptors is highly tunable (Zoltowski et al, 2009), and although in general the aim is to increase the sensitivity of the receptors, in this case its activation in the light should rather be dampened. Another approach is to apply an optogenetic system that uses light of other wavelengths, for example, far red light (Reichhart et al, 2016). For transgenic mice that express optogenetic receptors in light-exposed tissues, cages with colour-filtering properties should be used because reduction of ambient light in animal colonies is not compatible with animal welfare regulations in many countries.

Taken together, our study identified major strengths, but also limitations of optogenetic FGFR activation, which are likely dependent on the cell type and the genetic strategy. This should be taken into consideration in future in vitro and in vivo studies. Further optimization of optogenetic approaches for the activation of growth factor signaling will be important and should be tailored to the type of receptor and the target cell.

# Materials and Methods

### Recombinant proteins, antibodies, and primers

The following recombinant proteins were used: Murine EGF (E4127; Sigma-Aldrich), human FGF1 (100-17; PeproTech), human FGF2 (100-18; PeproTech), human FGF7 (100-19; PeproTech), and human FGF10 (100-26; PeproTech). Standard chemicals were from Sigma-Aldrich or Merck.

### Generation of OptoR2

The transgene encoding OptoR2 was generated by restriction digest and ligation. For this purpose, the intracellular domain (ICD) of human FGFR2 was tagged N-terminally with a myristoylation domain for membrane anchorage and fused with the aureochrome LOV domain from *Vaucheria frigida*, tagged at the C terminus with a hemagglutinin (HA) epitope. In the used region, mouse FGFR2 exhibits 92% sequence identity to human FGFR2, and the murine and human receptors can fully substitute each other's function (Ornitz & Itoh, 2015).

### Mice

The transgenic mouse line was generated by inserting the OptoR2-transgene under the control of the K14-promoter (Fig 2A) into C57BL/6 mouse oocytes by microinjection. After implanting the embryos into foster mice and delivery, their offspring was genotyped. Transgenic animals were used to establish three independent mouse lines, of which two exhibited undetectable transgene expression levels (Fig S2A). Mice were kept in C57BL/6 background under specific pathogen-free conditions according to federal guidelines with food and water ad libitum and a 12 h light–dark cycle. All mouse experiments were approved by the local veterinary authorities of Zurich, Switzerland (Kantonales Veterinäramt der Stadt Zürich, Switzerland). Mice with a single skin tumour that reached 1 cm diameter, with more than one tumour of at least 0.5-mm diameter or with tumours at a burdening body site were euthanized according to animal welfare regulations.

**List of primary antibodies.**

| Antigen | Host | Dilution | Application | Catalogue no. | Manufacturer |
|---|---|---|---|---|---|
| AKT | Rabbit | 1:1,000 | Western blot | 9272 | Cell Signaling Technologies |
| AKT (phospho T308) | Rabbit | 1:1,000 | Western blot | 9275 | Cell Signaling Technologies |
| AKT (phospho S473) | Rabbit | 1:1,000 | Western blot | 3787 | Cell Signaling Technologies |
| CD31 | Rat | 1:100 | Immunofluorescence | 553370 | BD Biosciences |
| E-Cadherin | Mouse | 1:100 | Immunofluorescence | 610181 | BD Biosciences |
| ERK | Rabbit | 1:1,000 | Western Blot | 9102 | Cell Signaling Technologies |
| ERK (phospho T202/Y204) | Rabbit | 1:100 | Immunofluorescence | 9101 | Cell Signaling Technologies |
| | | 1:1,000 | Western blot | | |
| FGFR2 (phospho Y769) | Rabbit | 1:1,000 | Western blot | PA5-105880 | Thermo Fisher Scientific |
| FRS2$\alpha$ (phospho Y196) | rabbit | 1:1.000 | Western blot | 3864 | Cell Signaling Technologies |
| FRS2$\alpha$ (phospho Y436) | Rabbit | 1:1,000 | Western blot | 3861 | Cell Signaling Technologies |
| GAPDH | Mouse | 1:5,000 | Western blot | 5G4cc | HyTest |
| HA (OptoR2) | Rabbit | 1:100 | Immunofluorescence | 3724 | Cell Signaling Technologies |
| | | 1:1,000 | Western blot | | |
| Ki67 | Rabbit | 1:500 | Immunohistochemistry, Immunofluorescence | ab16667 | Abcam |
| p38 | Rabbit | 1:1,000 | Western blot | 9212 | Cell Signaling Technologies |
| p38 (phospho T180/Y182) | Rabbit | 1:1,000 | Western blot | 9211 | Cell Signaling Technologies |
| PLC$\gamma$ (phospho S1248) | Rabbit | 1:1,000 | Western blot | 4510 | Cell Signaling Technologies |

For genotyping, skin obtained upon ear clipping was digested with proteinase K (A3830; AppliChem) and then used for the genotyping PCR. The genotyping mix contains 0.5 $\mu$l tissue lysate, 8 $\mu$l KAPA2G genotyping mix (KK5620; Roche), 6 $\mu$l ddH$_2$O, and 1 $\mu$l primer mix (10 $\mu$M). The PCR reaction was performed as described in the table, and the samples were analysed in a 2% agarose gel in SBA buffer containing 20 mM NaOH and 80.8 mM boric acid in water.

### Establishment of murine primary keratinocyte cultures

Primary keratinocytes were obtained from 3- to 5-d old pups as described previously (Braun et al, 2002), with few modifications. The pups were euthanized and the whole trunk skin was peeled off. After scraping off the subcutaneous fat tissue, the skin was incubated with 0.8% (wt/vol) trypsin (27250-018; Thermo Fisher Scientific) in DMEM (Merck) for 1 h at 37°C. Afterwards, the epidermis was separated from the dermis and incubated for 30 min at 37°C with 0.025% (wt/vol) DNase (DN25; Sigma-Aldrich) in DMEM + 10% (vol/vol) FBS (A3160802,

LOT 2166297; Thermo Fisher Scientific). The cell pellet was resuspended in keratinocyte growth medium (Braun et al, 2002) and the cells seeded in cell culture dishes pre-coated with collagen IV (C7521; Sigma-Aldrich) in PBS (2.5 $\mu$g/cm$^2$). They were cultivated for up to 1 wk.

### Cell culture

Human embryonic kidney T (HEK 293T) cells (Sigma-Aldrich), low-passage human immortalized keratinocytes (HaCaT cells) (Boukamp et al, 1988) (kindly provided by Prof. P. Boukamp, Leibniz Institute for Environmental and Medical Research), primary and spontaneously immortalized murine keratinocytes (established in this study), and human primary foreskin fibroblasts (kindly provided by Dr. Hans-Dietmar Beer, University of Zurich, Switzerland) were used. They were cultivated in DMEM + 10% FBS (HEK 293T, HaCaT, and human primary fibroblasts) or keratinocyte growth medium as described previously (Braun et al, 2002) (primary and immortalized murine keratinocytes) and

**List of secondary antibodies.**

| Antigen species | Coupled with | Dilution | Application | Catalogue no. | Manufacturer |
|---|---|---|---|---|---|
| Mouse | Cy2 | 1:200 | Immunofluorescence | 115-225-003 | Jackson ImmunoResearch |
| Mouse | HRP | 1:5,000 | Western blot | W4021 | Promega |
| Rabbit | Biotin | 1:500 | Immunohistochemistry | 111-065-003 | Jackson ImmunoResearch |
| Rabbit | Cy2 | 1:200 | Immunofluorescence | 111-225-144 | Jackson ImmunoResearch |
| Rabbit | Cy3 | 1:200 | Immunofluorescence | 111-165-003 | Jackson ImmunoResearch |
| Rabbit | HRP | 1:5,000 | Western blot | W4011 | Promega |
| Rat | Cy3 | 1:200 | Immunofluorescence | 712-165-153 | Jackson ImmunoResearch |

**List of PCR primers.**

| Gene | Sequence | Application |
|------|----------|-------------|
| DUSP6 (human) | 5′-GTT CTA CCT GGA AGG TGG CT-3′ | qRT-PCR |
| | 5′-AGT CCG TTG CAC TAT TGG GG-3′ | |
| FGFR2 (human) | 5′-AGC TGG GGT CGT TTC ATC TG-3′ | qRT-PCR |
| | 5′-TTG GTT GGT GGC TCT TCT GG-3′ | |
| HBEGF (human) | 5′-TTA GTC ATG CCC AAC TTC ACT TT-3′ | qRT-PCR |
| | 5′-ATC GTG GGG CTT CTC ATG TTT-3′ | |
| OptoR2 | 5′-TGA CCA GCA GCT TGG CAT AA-3′ | Genotyping PCR |
| | 5′-GCT CTG CAA ATG GCA CAA CA-3′ | |
| | 5′-GCT CTG CAA ATG GCA CAA CA-3′ | qRT-PCR |
| | 5′-TGA CCA GCA GCT TGG CAT AA-3′ | |
| RPL27 (human) | 5′-TCA CCT AAT GCC CAC AAG GTA-3′ | qRT-PCR |
| | 5′-CCA CTT GTT CTT GCC TGT CTT-3′ | |
| Rps29 (mouse) | 5′-GGT CAC CAG CAG CTC TAC TG-3′ | qRT-PCR |
| | 5′-GTC CAA CTT AAT GAA GCC TAT GTC C-3′ | |
| VEGF (human) | Hs_VEGFA_6_SG, QuantiTect Primer Assay (QT01682072) | qRT-PCR |

passaged twice weekly. All cell lines were routinely screened for mycoplasma and found negative.

For the generation of stable cell lines, lentivirus particles were generated in HEK 293T cells. Cells were transiently transfected with a pcDNA-based plasmid (V79020; Thermo Fisher Scientific) or a pInducer plasmid (Addgene; no. 44012) harbouring the OptoR2 transgene or empty vector as a control, together with the packaging plasmids psPAX2 (Addgene no. 12260) and pCMV-VSV-G (Addgene no. 8454) using Lipofectamine 2000 Transfection reagent (11668030; Thermo Fisher Scientific). After virus production for 48 h, the supernatant was collected, cleared by filtration through a 0.45-µm filter and stored at −80°C. Target cells were transduced by incubating them for 48 h with the lentivirus-containing supernatant diluted 1:10 in DMEM + 10% FBS and selected with 1.5 µg/ml puromycin (P8833; Sigma-Aldrich) (HEK 293T with pcDNA plasmid) or 1 mg/ml G418 (11811031; Thermo Fisher Scientific) (HaCaT cells or primary human fibroblasts with pInducer plasmid).

### Starvation and illumination

HEK 293T cells, primary human fibroblasts, and HaCaT cells were starved by washing twice with PBS and changing the medium to DMEM without FBS. For the primary fibroblasts or HaCaT cells harbouring the doxycycline (Dox)-inducible construct, 50 ng/ml Dox

(D1822; Sigma-Aldrich) or the same amount of DMSO (1.02952; Merck) were added. Murine primary and immortalized keratinocytes were washed twice with PBS, and their medium was changed to defined keratinocyte serum-free medium (10744019; Thermo Fisher Scientific) without supplements, but with penicillin/streptomycin (P0781; Merck) and 4.2 pg/ml cholera toxin (C8052; Sigma-Aldrich). The cells were cultivated under starvation conditions for 16 h (HEK 293T), 24 h (keratinocytes), or 48 h (fibroblasts), respectively.

Afterwards, they were treated with 10 ng/ml FGF7, FGF10, FGF1, FGF2, or EGF or illuminated with blue light for different time periods. For short-term continuous illumination (up to 3 h), cell plates were put underneath a layer of LEDs (5119661; Light & More), while for long-term illumination, a light chamber for three six-well plates (Greiner) with one LED for each well was custom-built using an Arduino UNO board (Arduino). The plates were illuminated using "1 min on/15 min off" or "1 min on/60 min off" cycles or using continuous illumination. The respective dark control cells were covered and kept in the same incubator. Light intensities were measured with a Powermeter (LP1; Sanwa) and were around 27 and 0.2 µW/cm² in the light and in the dark, respectively.

**PCR protocol for genotyping.**

| Cycles | Time | Temperature |
|--------|------|-------------|
| 1× | 5 min | 95°C |
| 33× | 35 s | 95°C |
| | 35 s | 60°C |
| | 50 s | 72°C |
| 1× | 10 min | 72°C |
| Store at 8°C | | |

**qPCR protocol.**

| Cycles | Time | Temperature |
|--------|------|-------------|
| 1× | 10 min | 95°C |
| 50× | 10 s | 95°C |
| | 20 s | 60°C |
| | 20 s | 72°C |
| 1× | 5 s | 95°C |
| 1× | 1 min | 65°C |
| 1× | Slow heat | 95°C |
| Store at 40°C | | |

**Protocol for H&E staining.**

| Cycles | Solution | Time |
|---|---|---|
| 1× | Mayer's Hematoxylin solution | 3 min |
| 3× | ddH$_2$O | 10 s |
| 1× | Scott water | 30 s |
| 1× | ddH$_2$O | 10 s |
| 1× | 70% (v/v) ethanol/ddH$_2$O | 10 s |
| 1× | Eosin solution | 1 min |
| 2× | 80% (v/v) ethanol/ddH$_2$O | 10 s |
| 2× | 95% (v/v) ethanol/ddH$_2$O | 10 s |
| 2× | Ethanol | 10 s |
| 2× | Xylene | 10 min |
| Slides were finally mounted with Eukitt and air-dried | | |

### Scratch assay

Cells were cultivated on dishes pre-coated with collagen IV in PBS (2.5 $\mu$g/cm$^2$) until they reached full confluency, starved as described before and treated for 2 h with 2 $\mu$g/ml mitomycin C (M4287; Sigma-Aldrich) to block proliferation. Afterwards, a scratch was induced with a pipet tip and the cells were washed with PBS before starvation or treatment media were added. Pictures were taken at the same spot every 24 h for 72 h. They were analysed manually or with the MRI wound healing tool (https://github.com/MontpellierRessourcesImagerie/imagej_macros_and_scripts/wiki/Wound-Healing-Tool, accessed May 25, 2020).

### RNA isolation and qRT-PCR

Cells were washed with PBS before extracting their RNA with the Mini Total RNA Kit (IB47300; IBI Scientific) according to the manufacturer's instructions. 1 $\mu$g RNA was used for reverse transcription in a reaction volume of 20 $\mu$l with the iScript cDNA synthesis kit (1708890; Bio-Rad). The product was diluted 1:10 in water, and 5 $\mu$l cDNA were mixed with 5.5 $\mu$l LightCycler SYBR green (04887352001; Roche) and 0.5 $\mu$l primer mix (10 $\mu$M). The qPCR reaction was performed according to the program specified in the table, and data were analysed using the double $\delta$ cycle threshold (CT) method.

### Preparation of protein lysates and Western blot

Cells were washed with PBS, lysed in lysis buffer containing 240 mM Tris–HCl pH 6.8, 280 mM SDS, and 40% (vol/vol) glycerol, and heated to 95°C. After heating the samples for 10 min at 95°C and short centrifugation, their protein concentration was determined using the BCA Protein assay kit (23225; Thermo Fisher Scientific) and a GloMax Microplate Reader (Promega). Afterwards, the samples were diluted to 1 $\mu$g/$\mu$l using lysis buffer with a final concentration of 10 mM DTT and bromophenol blue.

Samples of 20 $\mu$g protein were separated by SDS–PAGE and transferred to Protran nitrocellulose membranes (10600001; GE Healthcare). For probing with phospho-antibodies, membranes were blocked for 1 h with 5% (wt/vol) BSA (P06-1391100; PAN Biotech) in Tris-buffered saline with Tween 20 (TBS-T) containing 25 mM Tris base, 137 mM NaCl, 2.7 mM KCl, and 0.1% (vol/vol) Tween 20, pH adjusted to 8.0, whereas the membranes for incubation with non-phospho antibodies were blocked for 1 h with 5% (wt/vol) skim milk powder (150141000000; Rapilait) in TBS-T. They were then incubated with the primary antibody diluted in 5% (wt/vol) BSA in TBS-T overnight at 4°C, washed thrice for 5 min with TBS-T, incubated for 1 h with the secondary antibody diluted in 5% (wt/vol) skim milk powder in TBS-T, and washed thrice for 10 min with TBS-T. Signals were developed using WesternBright Sirius HRP substrate (K-12043; Advansta) and the FUSION SOLO 6S chemiluminescence imaging system (Witec).

### Immunofluorescence staining of cells

Cells were cultivated on cover slips, washed with PBS, and fixed for 20 min with 4% (wt/vol) PFA (P6148; Sigma-Aldrich) in PBS. After washing with PBS, they were permeabilised for 5 min with 0.5% (vol/vol) Triton X-100 in PBS and washed twice with PBS. Epitopes were blocked for 1 h with 2% (wt/vol) BSA and 0.05% (vol/vol) Triton X-100 in PBS. Then, the cells were incubated for 1 h with the primary antibodies diluted in the blocking solution, washed thrice for 5 min with PBS, and incubated for 1 h with the secondary antibodies and Hoechst 33342 (1:3,000) diluted in 0.05% (vol/vol) Triton X-100 in PBS. After washing thrice for 10 min with PBS, they were post-fixed for 5 min with 1% (wt/vol) PFA in PBS, washed with PBS, and mounted with ProLong Gold Antifade Mountant (P10144; Thermo Fisher Scientific). Photomicrographs were taken with an Axiovert microscope (Carl Zeiss) equipped with a Axiocam 506 mono camera (Carl Zeiss) at 10× and 20× magnification or with a confocal microscope (Leica SP8 from Leica) at 63× magnification.

### H&E staining of tissue sections

Samples from normal skin or skin tumours were fixed with 4% (wt/vol) PFA in PBS overnight at 4°C and then gradually dehydrated using an increasing ethanol gradient before embedding in paraffin (39601006; Leica Biosystems). Sections (7 $\mu$m) were dewaxed and rehydrated with a decreasing ethanol gradient, before staining them with hematoxylin and eosin (H&E) according to the program specified in the table.

### Immunofluorescence and immunohistochemistry staining of tissue sections

PFA–fixed and paraffin-embedded tissue sections were rehydrated and treated for 40 min with 0.33% (vol/vol) H$_2$O$_2$ in methanol. After washing the sections twice for 10 min with PBS, antigens were unmasked in 10 mM sodium citrate pH 6 (71405; Fluka) for 1 h at 95°C. After washing twice for 10 min with PBS, sections were blocked for 1 h with 12% (wt/vol) BSA in PBS and incubated overnight at 4°C with the primary antibodies diluted in blocking solution.

For immunofluorescence staining, the sections were then washed twice for 10 min with PBS, incubated for 1 h with the secondary antibodies diluted in PBS, before washing them thrice for 10 min with PBS. They were post-fixed for 5 min with 1% (wt/vol) PFA in PBS, washed with PBS and mounted with ProLong Gold Antifade Mountant. Photomicrographs were taken with an Axiovert microscope (Carl Zeiss) equipped with an Axiocam 506 mono camera (Carl Zeiss) at 10× magnification.

For immunohistochemistry, sections were washed thrice for 10 min with PBS, incubated for 45 min with the secondary antibody diluted in PBS, before washing them thrice for 5 min with PBS. Afterwards, they were incubated with the VECTASTAIN Elite ABC Kit Peroxidase solution (PK-4000; Maravai Life Sciences), washed again thrice for 5 min with PBS, before diaminobenzidine Kit solution (SK-4100; Maravai Life Sciences) was added. They were incubated until the brown staining was visible. The reaction was stopped by submersing the slides in tap water. Sections were counterstained with hematoxylin and mounted with Mowiol (81381; Sigma-Aldrich). Photomicrographs were taken using a light microscope (Axioskop 2; Carl Zeiss) equipped with a Axiocam 512 colour camera (Carl Zeiss) at 2.5×, 10× and 20× magnification.

### Statistical analysis

Statistical analysis was performed using the Prism 8 software (GraphPad Software Inc.). Quantitative data are presented as mean ± SEM. Significance was calculated using Mann–Whitney U test. $*P \leq 0.05$; $**P \leq 0.01$; $***P \leq 0.001$; $****P \leq 0.0001$.

## Supplementary Information

## Acknowledgements

We thank Connor Richterich and Patricia Reinert, ETH Zurich, for invaluable experimental help; Manuela Pérez Berlanga, University Zurich, for help with the confocal imaging; Lukas Fischer for help with electrical engineering; Thomas Hennek, Sol Taguinod, and Dr. Stephan Sonntag, EPIC Phenomics Center, ETH Zürich, for the generation and maintenance of K14-OptoR2 mice; and Dr. Petra Boukamp, Leibniz Institute, Düsseldorf, Germany, for early-passage HaCaT keratinocytes. This work was supported by the ETH Zurich (grant ETH-06 15-1 to S Werner and L Maddaluno), the Swiss National Science Foundation (grant 31003B-189364 to S Werner), and a Marie Curie postdoctoral fellowship from the European Union (to L Maddaluno).

## Author Contributions

T Rauschendorfer: conceptualization, formal analysis, investigation, visualization, methodology, and writing—original draft.
S Gurri: data curation, formal analysis, and investigation.
I Heggli: formal analysis, validation, and investigation.
L Maddaluno: conceptualization, funding acquisition, investigation, and methodology.
M Meyer: formal analysis and investigation.
A Ingles-Prieto: investigation and methodology.
H Janovjak: conceptualization, resources, methodology, and writing—review and editing.
S Werner: conceptualization, resources, supervision, funding acquisition, project administration, and writing—review and editing.

### Conflict of Interest Statement

The authors declare that they have no conflict of interest.

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
