## [Reviewer comments · Life Science Alliance]

Life Science Alliance

Acute and chronic effects of a light-activated FGF receptor in keratinocytes in vitro and in mice

Sabine Werner, Theresa Rauschendorfer, Selina Gurri, Irina Heggli, Luigi Maddaluno, Michael Meyer, Álvaro Inglés-Prieto, and Harald Janovjak

DOI: <https://doi.org/10.26508/lsa.202101100>

Corresponding author(s): Sabine Werner, ETH Zurich and Harald Janovjak, Monash University

Review Timeline:

Submission Date:	2021-04-17
Editorial Decision:	2021-06-15
Revision Received:	2021-08-17
Editorial Decision:	2021-09-02
Revision Received:	2021-09-06
Accepted:	2021-09-07

Transaction Report:

June 15, 2021

Re: Life Science Alliance manuscript #LSA-2021-01100-T

Prof. Sabine Werner
Institute of Molecular Health Sciences, ETH Zurich
Department of Biology
Otto-Stern Weg 7, HPL F12
Zurich, Zurich 8093
Switzerland

Dear Dr. Werner,

Thank you for submitting your manuscript entitled "Acute and chronic effects of a light-activated FGF receptor in keratinocytes in vitro and in mice" to Life Science Alliance. The manuscript was assessed by expert reviewers, whose comments are appended to this letter. We invite you to submit a revised manuscript addressing the Reviewer comments.

Thank you for this interesting contribution to Life Science Alliance. We are looking forward to receiving your revised manuscript.

Sincerely,

Eric Sawey, PhD
Executive Editor
Life Science Alliance

- A letter addressing the reviewers' comments point by point.
- An editable version of the final text (.DOC or .DOCX) is needed for copyediting (no PDFs).
- High-resolution figure, supplementary figure and video files uploaded as individual files: See our detailed guidelines for preparing your production-ready images, <https://www.life-science-alliance.org/authors>
- Summary blurb (enter in submission system): A short text summarizing in a single sentence the study (max. 200 characters including spaces). This text is used in conjunction with the titles of papers, hence should be informative and complementary to the title and running title. It should describe the context and significance of the findings for a general readership; it should be written in the present tense and refer to the work in the third person. Author names should not be mentioned.

B. MANUSCRIPT ORGANIZATION AND FORMATTING:

Reviewer #1 (Comments to the Authors (Required)):

Rauschendorfer et al. describes the effects of a light-activated FGF receptor in keratinocytes both in vitro and in mice, highlighting the advantages of this approach, but also its current limitations. This study presents a novel and useful tool to study cell signalling - in this case FGFR2b signalling - and could therefore generate great interest in the signalling community. However, the authors limit their analysis to two cell types only, of which HaCaT expresses also endogenous FGFR2b. It is therefore not entirely clear how broad this optogenetic tool might become.

Major points:

1) The analysis of signalling activation is mainly based on ERK phosphorylation. What happens to FGFR2b phosphorylation itself? The authors should use available antibodies against

phosphorylated FGFR to check whether the kinetic of FGFR activation is similar between OptoR2 and either the endogenous FGFR2b (in HaCaT) or another FGFR2b construct (in HEK 293T). A comparison with FGF7 and FGF10 would be interesting as well.

2) What about the activation of PLCgamma and the recruitment of known FGFR2 effectors like FRS2 or PLCgamma to OptoR2? Can a different set of binding partners explain the differences in receptor stability and regulation?

3) Can the authors speculate more about why chronic FGFR2b signalling in particular is not tolerated by cells?

4) Ideally, the authors should test their construct in a third different cell type to fully demonstrate their hypothesis that this optogenetic construct works differently in different cell types.

5) Is it possible that the presence of endogenous FGFR2b in keratinocytes and in HaCaT affect the results presented here?

Minor points:

1) Can the authors explain what they mean with "non-neuronal" optogenetics?

2) How do the authors explain the difference in signal intensity between light-activate OptoR2 and FGF7-induced FGFR2b?

Reviewer #2 (Comments to the Authors (Required)):

In this manuscript Rauschendorfer and co-workers developed a light-sensitive version of the FGFR-2 (OptoR2) by adapting a previously published strategy to photo-tag FGF-R. It consists in tagging the intracellular region of the receptor with a photosensitive LOV domain and removal of the extracellular domain which is required for ligand binding. Using 293T cells they demonstrate convincingly that upon blue light stimulation OptoR2 is capable of activating downstream signalling of comparable strength as endogenous ligands. They next generated a transgenic mouse line expressing OptoR2 under the keratin promoter and characterise the efficacy of this approach to control FGF-R signalling in vivo. This is the most relevant part of the study which will be highly informative for a wide audience. They report that primary keratinocytes isolated from transgenic lines initially respond to light but over time they become light-insensitive. They demonstrate that is due to a combination of RNA and protein downregulation due to low but constant light stimulation of the adult transgenic mice that can not be kept in the dark (as it can be done with developing fly embryos) for ethical reasons. Consistently they show that some transgenic mice develop tumors specifically in area of the skin that is more exposed to light. They further developed an inducible transgenic cell line expressing OptoR2 under the control of doxycycline and demonstrate that also under this condition continuous light stimulation for 3h. induces receptor downregulation. Overall this study is well conducted and provides useful information for research groups aiming to implement optogenetics to control signalling in mice and other vertebrate systems.

I suggest to expand the discussion and provide some suggestions based on the author's experience on what could be tried next (e.g different promoters, modification of the 3'UTR photoreceptors with different light sensitivity etc.). Maybe a summary table suggesting do and don't would be useful.

Reviewer #3 (Comments to the Authors (Required)):

In this report, the authors aimed to characterise the use of an optogenetically activated FGFR2

construct, in order to devise tools for controlled activation of this signalling receptor in vitro and in vivo. The study measures the dynamics of receptor activation in two settings cultured cells and transgenic mice. A very interesting set of observations have been reported, in particular about the negative dynamics of receptor activation. I.e. Long term downregulation of or some non-responsiveness of cells to the introduced constructs. Some work on subcellular trafficking of OptoR molecules would have been useful to shed light on this angle, but this is elegantly suggested as the future work. Overall the studies have been carried out meticulously and provide a basis for wider investigations. They also provide the cell signalling community with valuable tools for controlled induction of receptor activation.

A few queries/ suggestions for improvement of the manuscript.

1. May be better to have a schematic/map of the construct used to generate the transgenic mice, to showing the key regulatory features. This will help others to perhaps learn and potentially optimize the constructs further. Similarly, a schematic of the doxycycline response construct could be provided.
2. Do HaTaC cells have the same FGFR expression profile as HEK cells? For HEK cells this was clearly spelled out.
3. Where the transgenic mice subsequently maintained on C57BLk background or where the subsequent crosses on mixed genetic background? Genetic composition may be making a difference to level of transgene retention/ activation. Any further comments/ speculation of why males were more prone to skin tumor - if there is published literature in similar setting, this could be cited.

Eric Sawey, PhD
Executive Editor
Life Science Alliance

HPL F16.1
Otto-Stern-Weg 7
ETH Zurich
CH-8093 Zurich

Prof. Dr. Sabine Werner

Tel. +41 44 633 3941

sabine.werner@biol.ethz.ch
<http://www.mhs.biol.ethz.ch>

August 16, 2021

LSA-2021-01100-T

Dear Dr Sawey:

Thank you for your mail of June 15, 2021, with the comments of three reviewers on our manuscript entitled "Acute and chronic effects of a light-activated FGF receptor in keratinocytes *in vitro* and in mice".

We were pleased to see that all reviewers were generally positive about our manuscript, and we would like to thank them for their helpful and constructive suggestions and comments.

We have revised the manuscript based on these comments and our specific responses are listed below.

We hope that with these revisions our manuscript will be acceptable for publication in *Life Science Alliance*.

Thank you for considering the revised version of our manuscript.

Sincerely yours,

Sabine Werner

Reviewer #1 (Comments to the Authors (Required)):

Rauschendorfer et al. describes the effects of a light-activated FGF receptor in keratinocytes both in vitro and in mice, highlighting the advantages of this approach, but also its current limitations. This study presents a novel and useful tool to study cell signalling - in this case FGFR2b signalling - and could therefore generate great interest in the signalling community. However, the authors limit their analysis to two cell types only, of which HaCaT expresses also endogenous FGFR2b. It is therefore not entirely clear how broad this optogenetic tool might become.

Our response:

We thank the reviewer for the positive comments and for the helpful suggestions and criticisms.

Major points:

1) The analysis of signalling activation is mainly based on ERK phosphorylation. What happens to FGFR2b phosphorylation itself? The authors should use available antibodies against phosphorylated FGFR to check whether the kinetic of FGFR activation is similar between OptoR2 and either the endogenous FGFR2b (in HaCaT) or another FGFR2b construct (in HEK 293T). A comparison with FGF7 and FGF10 would be interesting as well.

Our response:

We agree that this would provide a useful addition. However, due to the very low copy number of FGFR2b in most cell types, including keratinocytes, detection of phospho-FGFR2b by Western blot remains challenging. Nevertheless, we tested two promising antibodies based on literature data and probed the samples shown in Fig 4C.

ThermoFisher, phospho-FGFR2 (Tyr769), PA5-105880: The result for the OptoR2 (75kDa) is now included in Fig 4C and the whole membranes are shown below. We indeed observed strong phosphorylation of the highly expressed OptoR2, demonstrating efficient activation of the recombinant receptor. We observed various bands of higher molecular weight, but there was no difference between non-treated and FGF7/FGF10-treated cells for these bands, although strong activation of ERK1/2 and other signaling effectors was seen in the same experiment, which demonstrates the appropriate activity of the FGFs. Therefore, we doubt that any of the high molecular weight bands represents endogenous pFGFR2 and that the affinity of the antibody may not be sufficient to detect endogenous pFGFR2 in total lysates.

ThermoFisher, phospho-FGFR2 (Ser782), PA5-64796: With this antibody we were also not able to observe any changes between non-treated and FGF-treated cells, and further not for light activation of OptoR2.

Taken together, we were unfortunately not able to detect autophosphorylation of endogenous FGFR2 in response to FGF treatment. However, phosphorylation of different signaling molecules that are known to be activated by FGFs was observed, including phosphorylation of pFGFR α , which is fairly specific for FGFR. In addition, FGF treatment induced the expression of known FGF target genes. Most importantly, autophosphorylation of OptoR2 was detected in response to light, further demonstrating the appropriate activation of this recombinant receptor.

2) *What about the activation of PLCgamma and the recruitment of known FGFR2 effectors like FRS2 or PLCgamma to OptoR2? Can a different set of binding partners explain the differences in receptor stability and regulation?*

Our response:

As shown in Figure 4D, FRS2 α was phosphorylated in response to activation of endogenous FGFR2b and also of OptoR2. A similar result has now been obtained for PLC- γ , and the Western blot data have been included (Fig 4C and S3B, revised manuscript). This result was verified with primary human fibroblasts.

3) *Can the authors speculate more about why chronic FGFR2b signalling in particular is not tolerated by cells?*

Our response:

We hypothesize that chronic FGFR2b signaling may have an effect on the differentiation of keratinocytes. This is supported by the delayed keratinocyte differentiation in transgenic mice expressing a dominant-negative mutant of FGFR2b in suprabasal keratinocytes (Werner et al., 1993) and by the induction of early and late differentiation in HaCaT keratinocytes by overexpression of FGFR2b and stimulation with FGF7

(Belleudi et al., 2011, Rosato et al., 2018). Therefore, it is possible that premature differentiation of cells expressing high levels of OptoR2 or being highly responsive to OptoR2 signaling are lost during the cultivation cycles, which ultimately leads to a loss of OptoR2 level/responsiveness over time. This had been mentioned in the combined Results and Discussion of the previous version, and it is now mentioned in detail in the separate Discussion of the revised manuscript (page 9, lines 18-24).

4) Ideally, the authors should test their construct in a third different cell type to fully demonstrate their hypothesis that this optogenetic construct works differently in different cell types.

Our response:

First, we would like to point out that we do not think that the optogenetic construct works differently in different cell types. In fact, our results demonstrate that the early signaling effects are very similar in HEK 293T cells and keratinocytes. We only observe rapid OptoR2 down-regulation in keratinocytes, which did not allow us to study long-term effects.

Nevertheless, we agree that testing OptoR2 in an additional cell type would be interesting, although the focus of our study was on keratinocytes. Therefore, we also tested OptoR2 in primary human fibroblasts, which endogenously mainly express FGFR1. Indeed, also in this cell type, OptoR2 illumination robustly activated the classical FGFR signaling pathways to a comparable extent or even stronger than FGF1 or FGF2. The result is now shown in Fig S3B.

5) Is it possible that the presence of endogenous FGFR2b in keratinocytes and in HaCaT affect the results presented here?

Our response:

Grusch et al. showed 2014 that Opto-FGFR1 does not interact with WT-FGFR1 because of the absence of the large dimerization domain in the former, engineered receptor. By analogy, it is very unlikely that the presence of endogenous FGFR2b affects our results. In fact, and as mentioned above, the early response of keratinocytes to optogenetic activation of FGFR2 was robust and highly efficient. We now discuss this important point (page 9, lines 6-8).

Minor points:

1) Can the authors explain what they mean with "non-neuronal" optogenetics?

Our response:

Non-neuronal optogenetics is a term used by several researchers in the field (for example Wang et al. 2020) to describe the application of optogenetics in cells other than neurons with a focus on the control of signaling activity. We have explained the term on page 4, lines 11-13, but we removed it from the Abstract to avoid confusion.

2) How do the authors explain the difference in signal intensity between light-activate OptoR2 and FGF7-induced FGFR2b?

Our response:

HEK 293T cells express extremely low levels of endogenous FGFR. This is shown in the qPCR data presented in Fig S1A. Therefore, the expression level of the receptors correlates with the ERK1/2 signal intensity upon stimulation with ligand. Consistent with this assumption, activation of ERK1/2 signaling in keratinocytes was similar upon stimulation of the endogenous receptor by its ligand FGF7 or of OptoR2 by light (Fig 4A, C, D).

Reviewer #2 (Comments to the Authors (Required)):

In this manuscript Rauschendorfer and co-workers developed a light-sensitive version of the FGFR-2 (OptoR2) by adapting a previously published strategy to photo-tag FGF-R. It consists in tagging the intracellular region of the receptor with a photosensitive LOV domain and removal of the extracellular domain which is required for ligand binding. Using 293T cells they demonstrate convincingly that upon blue light stimulation OptoR2 is capable of activating downstream signalling of comparable strength as endogenous ligands. They next generated a transgenic mouse line expressing OptoR2 under the keratin promoter and characterise the efficacy of this approach to control FGF-R signalling in vivo. This is the most relevant part of the study which will be highly informative for a wide audience. They report that primary keratinocytes isolated from transgenic lines initially respond to light but over time they become light-insensitive. They demonstrate that is due to a combination of RNA and protein downregulation due to low but constant light stimulation of the adult transgenic mice that can not be kept in the dark (as it can be done with developing fly embryos) for ethical reasons. Consistently they show that some transgenic mice develop tumors specifically in area of the skin that is more exposed to light. They further developed an inducible transgenic cell line expressing OptoR2 under the control of doxycycline and demonstrate that also under this condition continuous light stimulation for 3h. induces receptor downregulation. Overall this study is well conducted and provides useful information for research groups aiming to implement optogenetics to control signalling in mice and other vertebrate systems. I suggest to expand the discussion and provide some suggestions based on the author's experience on what could be tried next (e.g different promoters, modification of the 3'UTR photoreceptors with different light sensitivity etc.). Maybe a summary table suggesting do and don't would be useful.

Our response:

We thank the reviewer for the positive comments and the important suggestion. We have expanded the discussion accordingly. As a possible next step we mention the use of optimized pulsed light conditions (Hennemann et al., 2018), tuning of the OptoR2 lifetime (LOV domains are highly tunable; Zoltowski et al., 2009), use of a different promoter to avoid the down-regulation, or use a different Opto system with another activation spectrum (far red light) (Reichhart et al., 2016). This is now summarized in the Discussion (page 10, lines 6-14).

Reviewer #3 (Comments to the Authors (Required)):

In this report, the authors aimed to characterise the use of an optogenetically activated FGFR2 construct, in order to devise tools for controlled activation of this signalling receptor in vitro and in vivo. The study measures the dynamics of receptor activation in two settings cultured cells and transgenic mice. A very interesting set of observations have been reported, in particular about the negative dynamics of receptor activation. I.e. Long term downregulation of or some non-responsiveness of cells to the introduced constructs. Some work on subcellular trafficking of OptoR molecules would have been useful to shed light on this angle, but this is elegantly suggested as the future work. Overall the studies have been carried out meticulously and provide a basis for wider investigations. They also provide the cell signalling community with valuable tools for controlled induction of receptor activation.

Our response: We thank the reviewer for the positive comments and for the helpful suggestions.

A few queries/ suggestions for improvement of the manuscript.

1. May be better to have a schematic/map of the construct used to generate the transgenic mice, to showing the key regulatory features. This will help others to perhaps learn and potentially optimize the constructs further. Similarly, a schematic of the doxycycline response construct could be provided.

Our response:

We have included schematic representations of the constructs in Fig 2A and S3A.

2. Do HaTaC cells have the same FGFR expression profile as HEK cells? For HEK cells this was clearly spelled out.

Our response:

In Fig S1A we show that HEK cells express much less FGFR2 than HaCaT cells. While HEK 293T cells express very low levels of all FGFRs, HaCaT cells express FGFR2 and FGFR3, but not FGFR1 and FGFR4. However, we do not think that this is relevant for the response to OptoR2, because it was previously shown that endogenous FGFR1 does not affect the activity of OptoR1 (Grusch et al., 2014). We have clarified this point in the discussion (page 9, lines 6-8).

3. Where the transgenic mice subsequently maintained on C57BLk background or where the subsequent crosses on mixed genetic background? Genetic composition may be making a difference to level of transgene retention/ activation. Any further comments/ speculation of why males were more prone to skin tumor - if there is published literature in similar setting, this could be cited.

Our response:

The transgenic mice were generated and continuously maintained in pure C57BL/6 background. We have added this information to Materials and Methods (page 13, line 15).

We speculate that the skin tumors occur mainly in male mice because they frequently fight and scratch each other when they are not housed individually. These scratches might be the initial stimulus for tumor formation, which is a frequent phenomenon at sites of mechanical irritation. However, we have no experimental proof for this hypothesis, and therefore we prefer to avoid speculations.

References:

- HENNEMANN, J., IWASAKI, R. S., GRUND, T. N., DIENSTHUBER, R. P., RICHTER, F. & MÖGLICH, A. 2018. Optogenetic Control by Pulsed Illumination. *ChemBioChem*, 19, 1296-1304.
- REICHHART, E., INGLES-PRIETO, A., TICHY, A. M., MCKENZIE, C. & JANOVIK, H. 2016. A phytochrome sensory domain permits receptor activation by red light. *Angewandte Chemie International Edition*, 55, 6339-6342.
- ZOLTOWSKI, B. D., VACCARO, B. & CRANE, B. R. 2009. Mechanism-based tuning of a LOV domain photoreceptor. *Nat Chem Biol*, 5, 827-34.

September 2, 2021

RE: Life Science Alliance Manuscript #LSA-2021-01100-TR

Prof. Sabine Werner
Institute of Molecular Health Sciences, ETH Zurich
Department of Biology
Otto-Stern Weg 7, HPL F12
Zurich, Zurich 8093
Switzerland

Dear Dr. Werner,

Thank you for submitting your revised manuscript entitled "Acute and chronic effects of a light-activated FGF receptor in keratinocytes in vitro and in mice". We would be happy to publish your paper in Life Science Alliance pending final revisions necessary to meet our formatting guidelines.

- please add ORCID ID for secondary corresponding author-he should has received instructions on how to do so
- please use the [10 author names, et al.] format in your references (i.e. limit the author names to the first 10)
- LSA allows supplementary figures, but no EV Figures - please update a callout on page 13 accordingly

Figure checks:

- blots in Figure S2, ERK 1/2 are not continuous. Please indicate in the Figure Legend what the drawn line indicates.

LSA now encourages authors to provide a 30-60 second video where the study is briefly explained. We will use these videos on social media to promote the published paper and the presenting author. Corresponding or first-authors are welcome to submit the video. Please submit only one video per manuscript. The video can be emailed to contact@life-science-alliance.org

A. FINAL FILES:

B. MANUSCRIPT ORGANIZATION AND FORMATTING:

Sincerely,

Eric Sawey, PhD

Executive Editor
Life Science Alliance
<http://www.lsjournal.org>

Reviewer #1 (Comments to the Authors (Required)):

The authors have answered to all my query and the manuscript can now be published.

Reviewer #3 (Comments to the Authors (Required)):

The authors have prepared a good rebuttal to the points raised by reviewers and accordingly the manuscript is much improved and clearer. The paper describes the advantages as well as the limitations of the light activated FGF-receptor constructs and so this will prove as valuable source of reference for further investigations.

September 7, 2021

RE: Life Science Alliance Manuscript #LSA-2021-01100-TRR

Prof. Sabine Werner
ETH Zurich
Department of Biology, Institute of Molecular Health Sciences
Otto-Stern Weg 7, HPL F12
Zurich, Zurich 8093
Switzerland

Dear Dr. Werner,

Thank you for submitting your Research Article entitled "Acute and chronic effects of a light-activated FGF receptor in keratinocytes in vitro and in mice". It is a pleasure to let you know that your manuscript is now accepted for publication in Life Science Alliance. Congratulations on this interesting work.

DISTRIBUTION OF MATERIALS:

Again, congratulations on a very nice paper. I hope you found the review process to be constructive and are pleased with how the manuscript was handled editorially. We look forward to future exciting submissions from your lab.

Sincerely,
